# Exploring mechanical work changes in controlled ankle motion (CAM) boot walking: The effects of gait speed and shoe levelling

**Aaron Thomas**[1], **David E. Lunn**[1,2], **Josh Walker**[1]*

1 Carnegie School of Sport, Leeds Beckett University, Leeds, United Kingdom, 2 NIHR Leeds Biomedical Research Centre, Leeds Teaching Hospitals N.H.S. Trust, Leeds, United Kingdom

* josh.walker@leedsbeckett.ac.uk

## Abstract

Preferred walking speed (PWS) is lower when wearing a controlled ankle motion (CAM) boot, which can potentially make comparisons between footwear conditions difficult. Standardising walking speed accounts for this but lacks the ecological validity of PWS. The aim of this study was to compare acute biomechanical responses to CAM boot wear when walking is freely chosen and when it is controlled. Twelve healthy participants walked on an instrumented treadmill at their PWS and at three standardised speeds: 3, 4, and 5 km/h. They did so in three footwear conditions: (1) with a Rebound® Air Walker CAM boot on the right leg, (2) with a Rebound® Air Walker on the right leg and an Evenup Shoelift™ on the left, and (3) in normal footwear. Comparisons between footwear conditions were largely similar in the ipsilateral limb at PWS and at the standardised speeds, which included a decrease in total mechanical work and ankle joint work during CAM boot wear ($p < 0.001$). At the standardised speeds, total mechanical work and hip joint work were lower during CAM boot wear than wearing normal shoes and the Evenup Shoelift™ ($p \leq 0.014$), although there were no differences between footwear conditions at PWS ($p \geq 0.095$). As such, acute responses to CAM boot wear are different when speed is standardised compared to when speed is freely chosen, meaning conclusions cannot necessarily be transferred between approaches. Based on these differences observed between walking speeds, it would be prudent for future studies to try to maintain ecological validity by using PWS.

## Introduction

Controlled ankle motion (CAM) boots are a type of foot and ankle orthosis which are often prescribed to patients who have suffered trauma to the foot or ankle. For example, CAM boots are used in the early stages of rehabilitation following Achilles tendon (AT) rupture to restrict ankle motion and reduce anterior foot pressure [1,2], thus reducing AT load [3]. For AT rupture, the ankle joint is often fixed in a plantarflexed position during early-stage rehabilitation, whereas in later-stage rehabilitation, some CAM boot designs offer a controlled range of motion to modulate "weight-bearing". When CAM boots are used to treat diabetic foot ulcers, their primary purpose is to "offload" parts of the foot to aid with foot ulceration wound management [4].

**Data availability statement:** Anonymised data are available from the figshare database (DOI: https://doi.org/10.6084/m9.figshare.27959337. v1)

**Funding:** The author(s) received no specific funding for this work.

**Competing interests:** The authors have declared that no competing interests exist.

Regardless of the application or design of a CAM boot, its overall aim is to allow the patient to continue activities of daily living, such as walking or navigating up and down stairs, whilst also permitting rehabilitation examinations and exercises which would not be possible with conventional plaster casts. However, to continue activities of daily living, some compensatory mechanisms must occur to account for the lack of ankle joint contribution from the CAM boot-wearing limb. Firstly, unilateral CAM boot use has been shown to reduce a patient's preferred walking speed (PWS) [5,6], although this can be partially mitigated by the implementation of a shoe leveller, or "even-up walker" [7]. Even-up walkers are placed on the sole of the contralateral foot and are implemented to reduce, or offset, leg length discrepancies caused by the sole thickness of the CAM boot. In addition to mitigating the compromised gait speed, further biomechanical changes include alterations to various spatiotemporal, kinetic, and kinematic parameters [5,7–9]. Pelvic and thorax kinematic responses have also been reported previously [5], which might help explain the secondary site pain associated with CAM boot use [10]. However, there persists to be no clear explanation for why these biomechanical responses, and the associated reduction in PWS, might happen. It would be interesting to understand the underpinning factors that govern these responses, especially given they potentially lead to secondary site complications following prolonged use.

Given that CAM boots reduce a patient's PWS, it is not clear whether the biomechanical differences observed (e.g., spatiotemporal, joint kinetic parameters) are a direct impact of CAM boot wear or are caused by a change in walking speed. To combat this, standardising gait speed across all participants is possible, but lacks ecological validity and patients might adapt to a prescribed walking speed differently, thus altering normal function. Using PWS presents a different challenge when comparing across different participants, because walking speed alters spatiotemporal, kinematic, and kinetic gait parameters [11]. To explore how participants adapt during these different conditions, walking speed can be controlled across CAM boot and non-CAM boot conditions, and then compared with gait characteristics obtained at a PWS. Therefore, the aim of the current study was to investigate lower-limb gait patterns during CAM boot use when walking at a range of controlled walking speeds and PWS, as well as comparing these with normal footwear, with the primary null hypothesis being that controlling walking speed does not change our conclusions regarding biomechanical responses to CAM boot wear. This provides an important insight into the impact of CAM boots and whether trends are affected by controlling gait speed. The inclusion of an even-up walker in one footwear condition will permit comparison between CAM boot prescription with and without the consideration of accounting for leg length discrepancies. Given previous research found that the inclusion of shoe levellers increases PWS, it is plausible that they might also partially restore joint kinematic and kinetic function and changes in response to walking speed might differ. The results of this study could inform the importance of prescribing a shoe-leveller device for patients using a CAM boot.

## Materials and methods

### Participants

Twelve participants (eight males, four females; age: [mean ± S.D.] 29 ± 8 y; stature: 1.81 ± 0.86 m; body mass: 81.6 ± 13.7 kg) were recruited for this study. All participants were healthy and free of any musculoskeletal injury or neurological condition that might impact gait. Participants completed health screening and provided written informed consent prior to participation. The study was approved by the Local Research Ethics Committee (project number 66465), and was conducted in accordance with the Declaration of Helsinki [12].

## Data collection

Data collection took place between April and June 2022. Participants walked on a motorised, instrumented treadmill (Gaitway3D, h/p/cosmos, Germany), with a load cell located at each of the four corners to provide three-dimensional ground reaction forces at 2 kHz (Arsalis, Belgium). Participants were required to walk at three standardised speeds: 3, 4, and 5 km/h, as well as their own PWS. Walking speeds were chosen to cover speeds previously observed in participants when wearing a CAM boot [5,7,13,14]. PWS was determined using a series of overground gait trials, as described previously [7]. These speeds were analysed during three different footwear conditions: (1) wearing a CAM boot (Rebound® Air Walker, Össur, Iceland) on the right leg (BOOT); (2) wearing the CAM boot with an even-up walker (Evenup Shoelift™, Oped GmbH, Germany) on the contralateral left leg (EVEN); and (3) wearing their own trainers bilaterally to act a control condition with normal footwear (NORM). Therefore, 12 gait trials per participant were collected in total. The BOOT condition reflects what a patient who has suffered a lower-leg injury like AT rupture would be prescribed towards the end of rehabilitation, whilst EVEN includes the even-up walker for cases where this is offered to patients. In both BOOT and EVEN, the CAM boot was setup to create a "neutral", plantigrade ankle position (i.e., no plantarflexion or dorsiflexion), and no internal ankle range of motion (in any plane) was, in theory, permitted. Participants wore their own trainers on the left leg and the size of the CAM boot was determined by shoe size, following manufacturer recommendations. The first 2 minutes of each treadmill trial served as familiarisation, then five consecutive gait cycles (heel strike to ipsilateral heel strike) were recorded for analysis. Joint kinematics were collected at 250 Hz using a 14-camera optoelectronic system (Oqus 7+, Qualisys AB, Sweden). Retroreflective markers were placed according to the calibrated anatomical systems technique (CAST), where the locations of skin-based joint markers have been described previously [7,15]. The instrumented treadmill and motion capture system were synchronised via the digital integration of treadmill signals into the motion capture software, as per manufacturer recommendations. All footwear conditions were carried out in a randomised order, whilst walking speed was randomised within each footwear condition for each participant individually.

## Data processing

Ground reaction force data and the associated centre of pressure signals were filtered with a recursive, second-order (zero phase-lag), low-pass Butterworth filter with a cut-off frequency of 17.7 ± 3.6 Hz. Cut-off frequencies were determined individually for each signal using residual analyses [16] via a custom-written Matlab script (R2023a, MathWorks Inc., USA). Filtered kinetic data were then "split" into left-leg and right-leg signals using a signal decomposition algorithm [17,18]. Three-dimensional coordinates of the retroreflective markers were filtered with a recursive, second-order (zero phase-lag), low-pass Butterworth filter with a cut-off frequency of 6.0 Hz. Kinematic modelling was conducted in Visual3D (v6.01.36, C-Motion Inc., Canada) using six degrees-of-freedom, where the CAM boots were modelled as foot and shank segments. The mass of the boot was added to the foot segment during modelling. Kinetic and kinematic data were combined to provide joint kinetics, which were estimated using Inverse Dynamics. Joint work done was computed as the area under the joint moment-angular displacement curve, and was resolved in all three directions (sagittal, frontal, and transverse planes) for each lower-limb joint to estimate overall mechanical work done by each joint [19]:

$$W_{joint} = \left( \int M_X \, d\theta_X \right) + \left( \int M_Y \, d\theta_Y \right) + \left( \int M_Z \, d\theta_Z \right),$$

where $W_{joint}$ = joint mechanical work, $M$ represents joint moments, and $\theta$ represents joint angles in sagittal ($X$), frontal ($Y$), and transverse ($Z$) planes, respectively. Total mechanical work was defined as the sum of all lower-limb joint work [19]:

$$W_{tot} = W_{hip} + W_{knee} + W_{ankle},$$

where $W_{tot}$ = total mechanical work, and $W_{hip}$, $W_{knee}$, and $W_{ankle}$ represent work done by the hip, knee, and ankle joint, respectively. All work done values were normalised to body mass (J·kg$^{-1}$). Work done by each joint was also calculated as a proportion of their total contribution to $W_{tot}$ [7]. Spatiotemporal parameters were also computed in Visual3D for left and right legs individually during all speeds and conditions.

## Statistical analysis

Data are presented as mean ± standard deviation. All statistical analyses were conducted in SPSS (version 28, IBM, USA). Data were tested for normality prior to statistical analysis. A two-way analysis of variances (ANOVA) with repeated measures was used to test the main effects of walking speed and footwear condition on discrete, dependent variables, as well as speed × condition interaction effects. PWS trials were analysed independently using a one-way ANOVA with repeated measures. Significant main effects or interactions were explored using post-hoc tests with a Bonferroni adjustment. The right (ipsilateral) and left (contralateral) legs were analysed independently here, as the primary aim was to investigate within-limb effects of footwear condition and speed. Significance level was set at $p < 0.05$.

## Results

There was a significant main effect of condition on PWS ($F_{2,22} = 5.77$, $p = 0.010$), with NORM (4.9 ± 0.6 km/h) being significantly faster than BOOT (4.5 ± 0.7 km/h) ($p = 0.006$), but not EVEN (4.6 ± 0.3 km/h) ($p = 0.080$). BOOT and EVEN displayed similar PWS ($p = 1.000$). During the standardised speeds, average measured gait speed was always within 0.5% of the intended gait speed, with all speeds being significantly different to the other speeds ($F_{2,22} = 113293.03$, $p < 0.001$). There were no significant differences between conditions ($F_{2,22} = 1.63$, $p = 0.219$) and no speed × condition interaction ($F_{1.06,11.68} = 1.04$, $p = 0.371$).

In the ipsilateral leg, $W_{tot}$ was reduced in both BOOT ($p < 0.001$) and EVEN ($p \leq 0.002$) compared to NORM at PWS ($F_{2,22} = 23.47$, $p < 0.001$) (Fig 1B). No main effects of condition were found for $W_{tot}$ in the contralateral limb at PWS ($F_{1.06,11.68} = 3.04$, $p = 0.095$) (Fig 1A). At the standardised speeds, $W_{tot}$ increased with walking speed in the ipsilateral ($F_{1.12,12.30} = 131.83$, $p < 0.001$) and contralateral ($F_{2,22} = 165.36$, $p < 0.001$) legs, with all speeds being significantly different to the other two speeds ($p < 0.001$) (Fig 1A and 1B). There was also a main effect of condition in the ipsilateral limb for $W_{tot}$ ($F_{1.13,12.48} = 44.04$, $p < 0.001$), which was lower in BOOT and EVEN compared to NORM ($p < 0.001$) (Fig 1B). A condition × speed interaction was also found ($F_{1.32,14.56} = 7.33$, $p = 0.006$), but post-hoc testing indicated that all conditions responded to speed changes by increasing similarly. $W_{tot}$ significantly changed in the contralateral leg ($F_{2,22} = 13.18$, $p < 0.001$), which was lower in BOOT compared to NORM ($p = 0.047$) and EVEN ($p < 0.001$) (Fig 1A).

Ipsilateral $W_{ankle}$ at PWS was lower ($F_{1.06,11.68} = 76.75$, $p < 0.001$) in BOOT and EVEN compared to NORM ($p < 0.001$) (Fig 2B). There was also a main effect of condition for contralateral $W_{ankle}$ at PWS ($F_{2,22} = 4.41$, $p = 0.025$). Post-hoc testing showed that NORM $W_{ankle}$ was higher than BOOT ($p = 0.011$) but not EVEN ($p = 0.217$) (Fig 2A). At the standardised speeds, ipsilateral $W_{ankle}$ was significantly higher in NORM ($F_{1.13,12.48} = 169.19$, $p < 0.001$) compared to

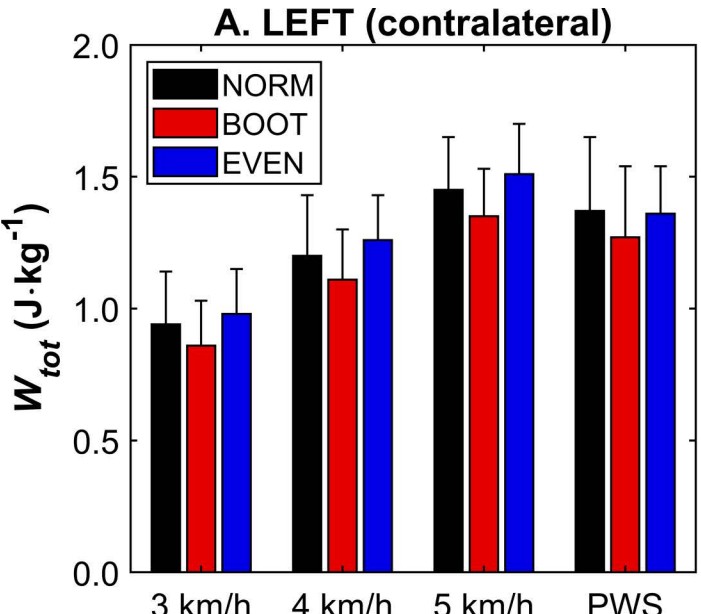
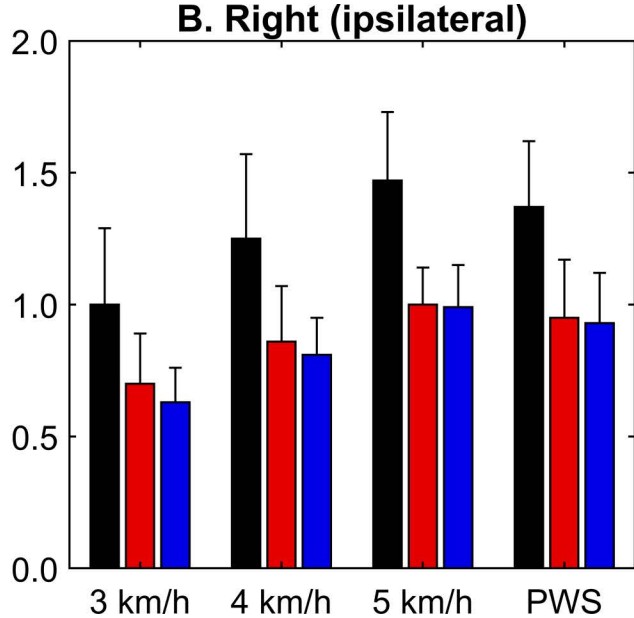

**Fig 1. Total mechanical work (*Wtot*) for each footwear condition at each speed in the contralateral (A) and ipsilateral (B) limb, showing that *Wtot* increases with gait speed in the standardised speed conditions in both limbs, but was also lower in BOOT and EVEN, compared with NORM, at all speeds in the ipsilateral limb.** $W_{tot}$ did not change between footwear conditions at PWS but was lower in BOOT at the standardised speeds. PWS = preferred walking speed. Data pertaining to this figure, along with a summary of statistical results, can be found in S1 Table.

BOOT and EVEN ($p < 0.001$) (Fig 2B). There was also a main effect of condition on contralateral leg ($F_{1.13,12.48} = 5.62$, $p = 0.029$). Post-hoc testing showed no individual differences ($p \geq 0.092$). Additionally, ipsilateral ($F_{2,22} = 45.62$, $p < 0.001$) and contralateral ($F_{1.12,12.30} = 42.78$, $p < 0.001$) $W_{ankle}$ both increased with walking speed (all post-hoc comparisons $p \leq 0.008$) (Fig 2A).

The relative ankle contribution to $W_{tot}$ was also lower in the ipsilateral limb ($F_{1.06,11.68} = 74.94$, $p < 0.001$) for BOOT and EVEN compared to NORM ($p < 0.001$) (Fig 2D), whilst contralateral relative ankle contribution was lower ($F_{1.06,11.68} = 7.33$, $p = 0.014$) in EVEN only ($p = 0.037$), at PWS (Fig 2C). At the standardised speeds, there was a main effect of speed for both ipsilateral ($F_{1.12,12.30} = 8.29$, $p = 0.012$) and contralateral ($F_{1.12,12.30} = 12.14$, $p = 0.003$) relative ankle contribution, which was lower at 5 km/h compared to 3 and 4 km/h in both limbs ($p \leq 0.032$), but not different between 3 and 4 km/h ($p \geq 0.129$) (Fig 2C and 2D). There was also a main effect of condition for both ipsilateral ($F_{1.13,12.48} = 527.32$, $p < 0.001$) and contralateral ($F_{2,22} = 29.98$, $p < 0.001$) limbs. In the ipsilateral limb, relative ankle contribution was higher in NORM than BOOT and EVEN ($p < 0.001$), whilst BOOT and EVEN were similar ($p = 1.000$) (Fig 2D). In the contralateral limb, relative ankle contribution was lower in EVEN than NORM and BOOT ($p \leq 0.001$), whilst NORM and BOOT were similar ($p = 0.131$) (Fig 2C). A significant speed × condition interaction effect was also observed for the contralateral limb, where the different conditions showed varying responses to speed (Fig 2C). NORM ankle relative contribution was lower at 5 km/h compared to 3 and km/h ($p \leq 0.049$), EVEN was lower at 5 km/h compared to 4 km/h only ($p = 0.004$), and all speeds were different to all other speeds in BOOT ($p \leq 0.007$).

At PWS, there was no main effect of condition on ipsilateral $W_{knee}$ ($F_{2,22} = 2.84$, $p = 0.080$), but there was in the contralateral limb ($F_{1.06,11.68} = 6.11$, $p = 0.018$), with BOOT displaying lower $W_{knee}$ compared to NORM ($p = 0.028$) and EVEN ($p = 0.019$) (Fig 3A and 3B). There was no main effect of condition on ipsilateral $W_{knee}$ during standardised speeds ($F_{1.13,12.48} = 3.91$,

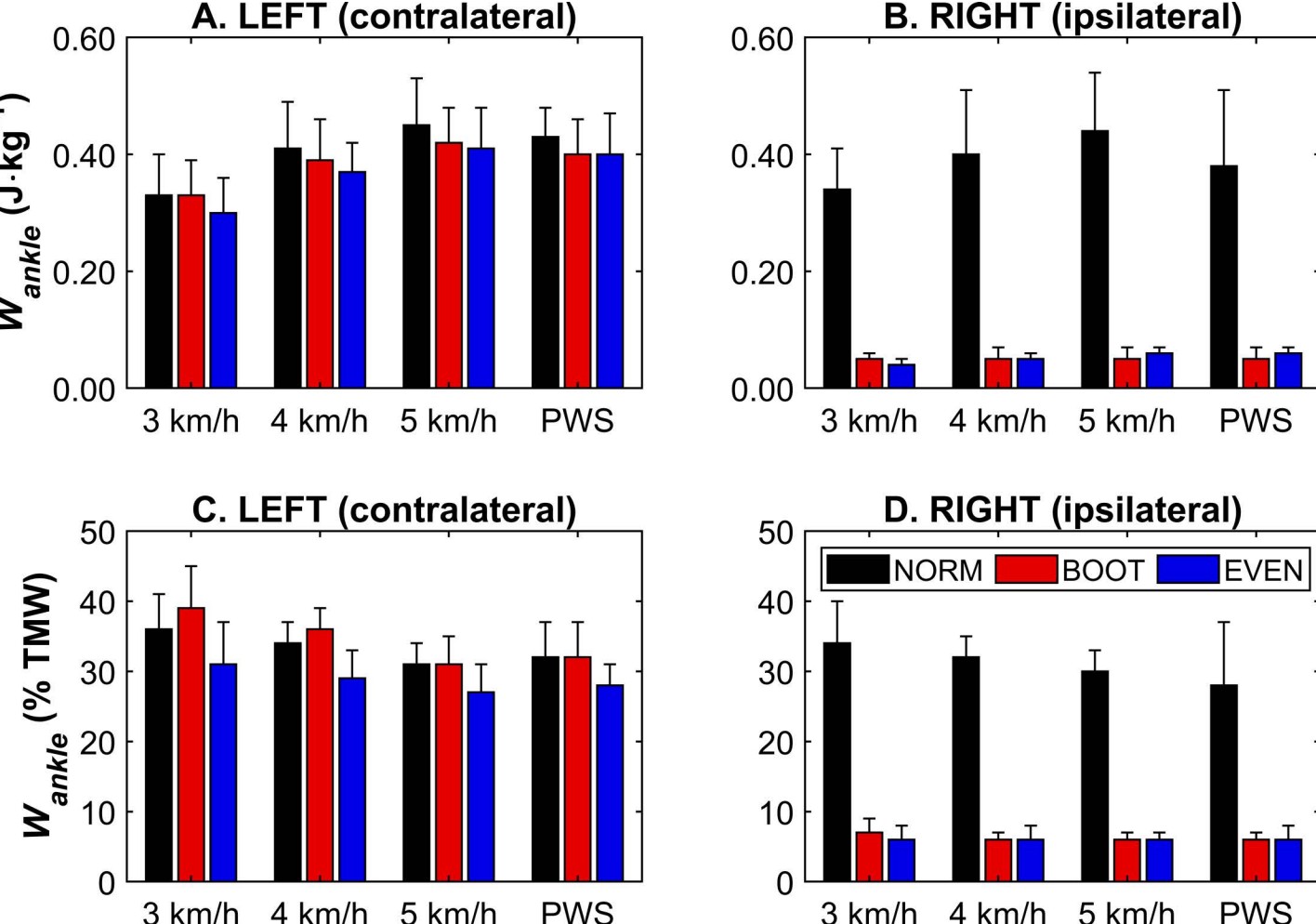

**Fig 2. Ankle joint mechanical work (*W*ankle) for each footwear condition at each speed, displayed relative to body mass in the contralateral (A) and ipsilateral (B) limb, and as a relative contribution to total mechanical work in the contralateral (C) and ipsilateral (D) limb.** Ipsilateral $W_{ankle}$ was lower in BOOT and EVEN, compared to NORM in all cases. Contralateral $W_{ankle}$ (relative to body mass) was lower in BOOT at PWS, but not in the standardised speeds, and increased with speed. Contralateral $W_{ankle}$ (relative contribution to total mechanical work) was lower in EVEN at PWS and at standardised speeds, as well as being lower at higher speeds. PWS = preferred walking speed. Data pertaining to this figure, along with a summary of statistical results, can be found in S1 Table.

$p = 0.064$) (Fig 3B), although there was a main effect of speed ($F_{1.12,12.30} = 86.34$, $p < 0.001$). Post-hoc testing showed that $W_{knee}$ increased with speed ($p < 0.001$) (Fig 3B). Main effects of speed were also found for $W_{knee}$ in the contralateral limb ($F_{2,22} = 82.56$, $p < 0.001$), with contralateral $W_{knee}$ increasing with speed ($p < 0.001$) (Fig 3A). There was also a main effect of condition on contralateral $W_{knee}$ ($F_{2,22} = 30.75$, $p < 0.001$), where $W_{knee}$ was higher in EVEN compared to NORM ($p = 0.009$) and BOOT ($p < 0.001$) (Fig 3A). BOOT also displayed lower $W_{knee}$ compared to NORM ($p = 0.002$) (Fig 3A).

At PWS and the standardised speeds, the ipsilateral knee joint's relative contribution to $W_{tot}$ was significantly lower ($F_{2,22} \geq 18.36$, $p < 0.001$) in NORM compared to BOOT and EVEN ($p \leq 0.003$), whilst BOOT and EVEN were similar ($p = 1.000$) (Fig 3D). In the contralateral limb, relative knee contribution at PWS was higher ($F_{2,22} = 6.69$, $p < 0.001$) in EVEN than BOOT ($p = 0.003$), but not NORM ($p = 0.176$). NORM and BOOT were also not different ($p = 0.088$) (Fig 3C). At the standardised speeds, contralateral relative knee contribution was also higher ($F_{2,22} = 31.24$, $p < 0.001$) in EVEN compared to NORM and BOOT ($p \leq 0.005$), whilst NORM

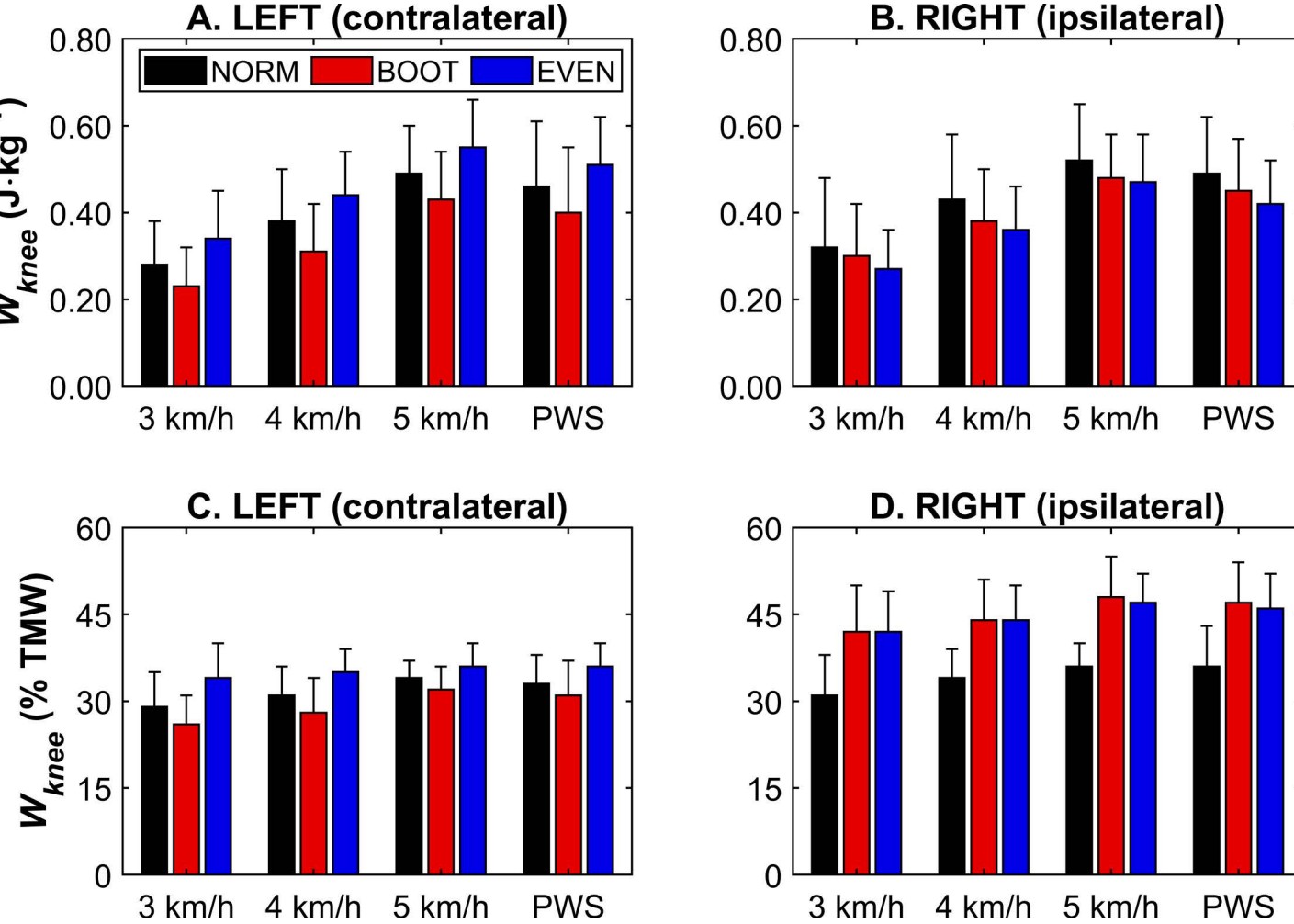

**Fig 3. Knee joint mechanical work (Wknee) for each footwear condition at each speed, displayed relative to body mass in the contralateral (A) and ipsilateral (B) limb, and as a relative contribution to total mechanical work in the contralateral (C) and ipsilateral (D) limb.** Ipsilateral $W_{knee}$ (relative to body mass) did not change between footwear conditions, but contralateral $W_{knee}$ (relative to body mass) was lower in BOOT at PWS, and higher in EVEN during the standardised speeds, and increased with speed. Ipsilateral $W_{knee}$ (relative contribution to total mechanical work) was lower in NORM at PWS and at standardised speeds, as well as increasing with speed. Contralateral $W_{knee}$ (relative contribution to total mechanical work) was mostly higher in EVEN and tended to increase with speed. PWS = preferred walking speed. Data pertaining to this figure, along with a summary of statistical results, can be found in S1 Table.

was higher than BOOT ($p < 0.001$) (Fig 3C). There was also a main effect of speed for both the ipsilateral ($F_{1.12,12.30} = 24.12$, $p < 0.001$) and contralateral ($F_{2,22} = 42.51$, $p < 0.001$), where relative knee contribution was higher at 5 km/h than at 3 and 4 km/h in both limbs ($p \leq 0.032$). There was no difference between 3 and 4 km/h in the contralateral limb ($p = 0.286$), but relative knee contribution was higher at 4 km/h than 3 km/h in the ipsilateral limb ($p = 0.042$) (Fig 3C,3D).

Ipsilateral and contralateral $W_{hip}$ both showed no main effect of condition ($F_{2,22; 1.06,11.68} \leq 1.29$, $p \geq 0.296$) at PWS (Fig 4A and 4B). However, there was a main effect of condition for contralateral $W_{hip}$ during the standardised speeds ($F_{2,22} = 5.21$, $p = 0.014$). Post-hoc testing showed that $W_{hip}$ during BOOT was significantly lower than during EVEN ($p = 0.032$) but not NORM ($p = 0.949$), with no significant differences between NORM and EVEN ($p = 0.098$) (Fig 4A). There were no main effects of condition for ipsilateral $W_{hip}$ at the standardised speeds ($F_{2,22} = 1.08$, $p = 0.357$) (Fig 4B). There were significant effects of speed on $W_{hip}$ in both

limbs ($F_{2,22} \geq 128.46$, $p < 0.001$), with post-hoc testing showing $W_{hip}$ significantly increased with speed in both limbs ($p < 0.001$).

At PWS, the hip joint's relative contribution to $W_{tot}$ was significantly lower ($F_{2,22} \geq 4.50$, $p \leq 0.023$) in NORM than BOOT and EVEN in the ipsilateral limb ($p < 0.001$), and BOOT only in the contralateral limb ($p = 0.005$). BOOT and EVEN were similar in both limbs ($p \geq 0.402$) (Fig 4C and 4D). At the standardised speeds, there was a main effect of condition for the ipsilateral limb ($F_{2,22} = 107.57$, $p < 0.001$), with NORM displaying lower relative hip contribution than BOOT and EVEN ($p < 0.001$), where BOOT and EVEN were similar ($p = 1.000$) (Fig 4D). There was also a main effect of speed ($F_{2,22} = 17.35$, $p < 0.001$), where relative hip contribution at 5 km/h was lower than 3 and 4 km/h ($p \leq 0.004$), where 3 and 4 km/h were similar ($p = 0.072$). There was also a significant speed × condition interaction effect, where NORM showed no individual effect of speed, whilst BOOT and EVEN showed a reduced relative hip

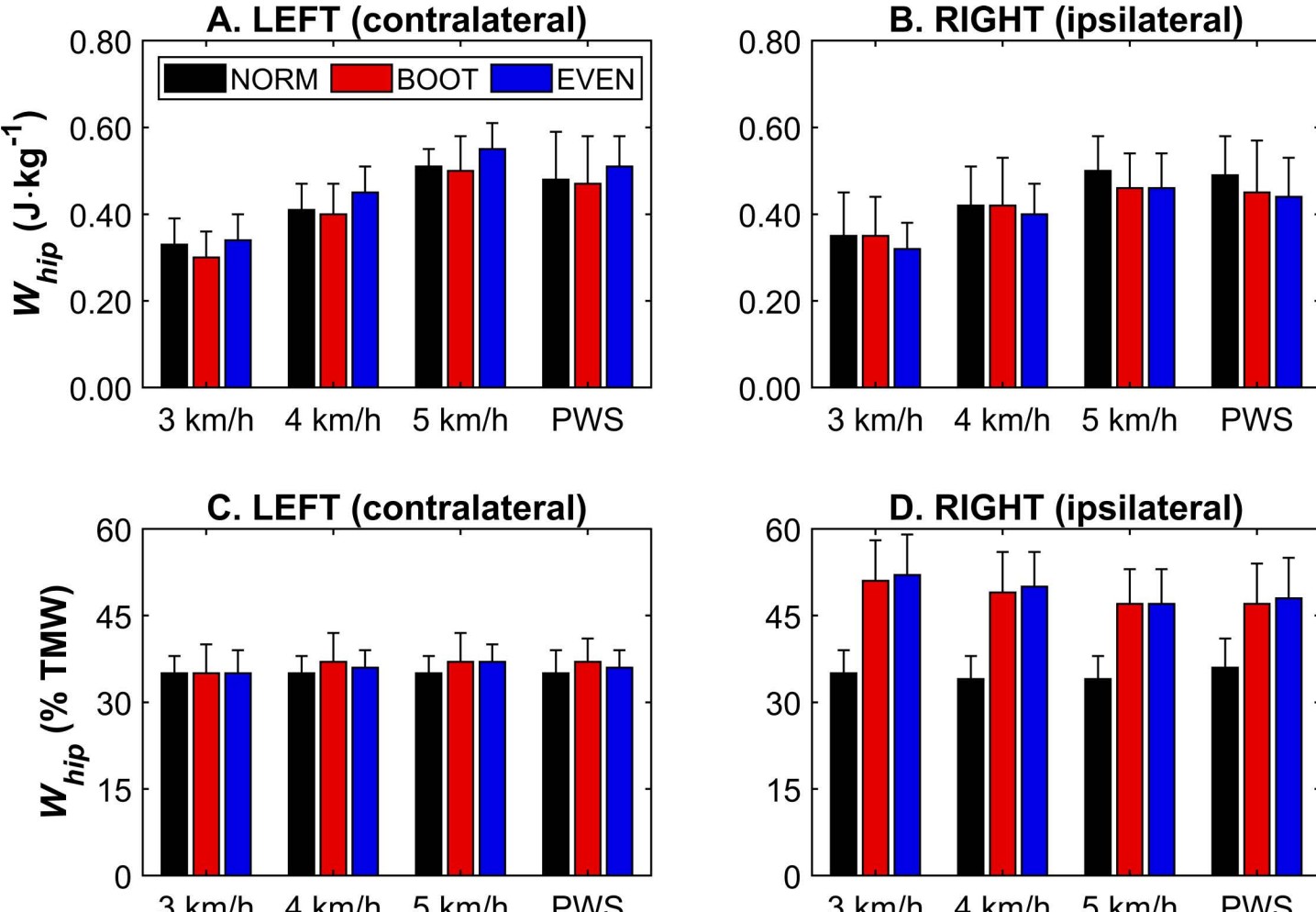

**Fig 4. Hip joint mechanical work ($Whip$) for each footwear condition at each speed, displayed relative to body mass in the contralateral (A) and ipsilateral (B) limb, and as a relative contribution to total mechanical work in the contralateral (C) and ipsilateral (D) limb.** $W_{hip}$ (relative to body mass) was similar between conditions but increased with speed. $W_{hip}$ (relative contribution to total mechanical work) was mostly lower in NORM at PWS and in the standardised speeds, although this was not the case for the contralateral limb at standardised speeds. Ipsilateral $W_{hip}$ (relative contribution to total mechanical work) was lower at higher speeds, but the contralateral limb did not change. PWS = preferred walking speed. Data pertaining to this figure, along with a summary of statistical results, can be found in S1 Table.

contribution at 5 km/h compared to the other two speeds ($p \leq 0.013$) (Fig 4D). There were no main effects of condition or speed, and no interaction effects, for the contralateral limb (Fig 4C).

Spatiotemporal parameters showed no main effects of condition at PWS, besides ipsilateral cadence ($F_{1.06,11.68} = 4.23$, $p = 0.048$) and swing time ($F_{1.06,11.68} = 9.78$, $p = 0.007$), and contralateral swing time ($F_{2,22} = 4.20$, $p = 0.028$) (Table 1). At the standardised speeds, there was also a

**Table 1. Spatiotemporal parameters for each footwear condition at preferred walking speed (PWS) and each of the standardised walking speeds. Speed influenced all spatiotemporal parameters reported, but footwear condition did not. Some main effects of condition were present at PWS but not in the standardised speeds, whilst some were present in the standardised speeds but not at PWS.**

| Parameter | Limb | Speed | NORM | BOOT | EVEN | ANOVA |
|---|---|---|---|---|---|---|
| **Step length (m)** | *Left**  | PWS | 0.73 ± 0.09 | 0.71 ± 0.08 | 0.72 ± 0.06 | N/A |
| | | 3 km/h<br>4 km/h<br>5 km/h | 0.58 ± 0.05<br>0.67 ± 0.05<br>0.75 ± 0.04 | 0.57 ± 0.04<br>0.66 ± 0.04<br>0.74 ± 0.06 | 0.58 ± 0.03<br>0.67 ± 0.03<br>0.77 ± 0.04 | **Condition:** BOOT <EVEN ($p = 0.005$)<br>**Speed:** All speeds different to all other speeds ($p < 0.001$) |
| | *Right* | PWS | 0.73 ± 0.08 | 0.72 ± 0.07 | 0.71 ± 0.06 | N/A |
| | | 3 km/h<br>4 km/h<br>5 km/h | 0.58 ± 0.07<br>0.66 ± 0.06<br>0.75 ± 0.05 | 0.58 ± 0.06<br>0.68 ± 0.06<br>0.76 ± 0.04 | 0.56 ± 0.04<br>0.67 ± 0.04<br>0.76 ± 0.05 | **Speed:** All speeds different to all other speeds ($p < 0.001$) |
| **Stance time (s)** | *Left**  | PWS | 0.692 ± 0.104 | 0.730 ± 0.089 | 0.741 ± 0.073 | N/A |
| | | 3 km/h<br>4 km/h<br>5 km/h | 0.872 ± 0.061<br>0.740 ± 0.038<br>0.670 ± 0.039 | 0.862 ± 0.042<br>0.740 ± 0.042<br>0.691 ± 0.073 | 0.884 ± 0.053<br>0.779 ± 0.035<br>0.697 ± 0.031 | **Condition:** BOOT <EVEN ($p = 0.030$)<br>**Speed:** All speeds different to all other speed ($p < 0.001$) |
| | *Right* | PWS | 0.682 ± 0.097 | 0.708 ± 0.081 | 0.709 ± 0.086 | N/A |
| | | 3 km/h<br>4 km/h<br>5 km/h | 0.865 ± 0.057<br>0.731 ± 0.039<br>0.668 ± 0.039 | 0.848 ± 0.045<br>0.730 ± 0.025<br>0.670 ± 0.064 | 0.867 ± 0.042<br>0.738 ± 0.032<br>0.656 ± 0.028 | **Speed:** All speeds different to all other speed ($p < 0.001$) |
| **Swing time (s)** | *Left**  | PWS | 0.420 ± 0.037 | 0.428 ± 0.031 | 0.418 ± 0.033 | **Condition:** EVEN <BOOT ($p = 0.034$) |
| | | 3 km/h<br>4 km/h<br>5 km/h | 0.482 ± 0.048<br>0.437 ± 0.031<br>0.412 ± 0.029 | 0.481 ± 0.034<br>0.440 ± 0.024<br>0.417 ± 0.027 | 0.475 ± 0.044<br>0.427 ± 0.031<br>0.404 ± 0.030 | **Speed:** All speeds different to all other speed ($p < 0.001$) |
| | *Right* | PWS | 0.430 ± 0.043 | 0.450 ± 0.037 | 0.453 ± 0.034 | **Condition:** NORM <BOOT ($p = 0.003$) & EVEN ($p = 0.031$) |
| | | 3 km/h<br>4 km/h<br>5 km/h | 0.489 ± 0.054<br>0.434 ± 0.034<br>0.417 ± 0.030 | 0.495 ± 0.034<br>0.459 ± 0.036<br>0.440 ± 0.034 | 0.504 ± 0.038<br>0.470 ± 0.036<br>0.445 ± 0.034 | **Condition:** NORM <BOOT ($p = 0.028$) & EVEN ($p = 0.007$)<br>**Speed:** All speeds different to all other speed ($p < 0.001$)<br>**Interaction:** Condition reacted to speeds similarly ($p \leq 0.048$) |
| **Cycle time (s)** | *Left* | PWS | 1.111 ± 0.133 | 1.158 ± 0.110 | 1.159 ± 0.098 | N/A |
| | | 3 km/h<br>4 km/h<br>5 km/h | 1.354 ± 0.106<br>1.177 ± 0.067<br>1.083 ± 0.064 | 1.343 ± 0.072<br>1.180 ± 0.060<br>1.108 ± 0.090 | 1.371 ± 0.072<br>1.208 ± 0.060<br>1.101 ± 0.056 | **Speed:** All speeds different to all other speed ($p < 0.001$) |
| | *Right* | PWS | 1.111 ± 0.133 | 1.159 ± 0.110 | 1.162 ± 0.099 | N/A |
| | | 3 km/h<br>4 km/h<br>5 km/h | 1.354 ± 0.107<br>1.180 ± 0.069<br>1.085 ± 0.064 | 1.343 ± 0.073<br>1.189 ± 0.053<br>1.110 ± 0.089 | 1.371 ± 0.072<br>1.208 ± 0.060<br>1.102 ± 0.056 | **Speed:** All speeds different to all other speed ($p < 0.001$) |
| **Cadence (Hz)** | *Left* | PWS | 1.86 ± 0.21 | 1.76 ± 0.14 | 1.78 ± 0.15 | N/A |
| | | 3 km/h<br>4 km/h<br>5 km/h | 1.50 ± 0.12<br>1.73 ± 0.12<br>1.86 ± 0.11 | 1.50 ± 0.09<br>1.73 ± 0.14<br>1.83 ± 0.12 | 1.49 ± 0.49<br>1.71 ± 0.09<br>1.87 ± 0.10 | **Speed:** All speeds different to all other speed ($p < 0.001$) |
| | *Right** | PWS | 1.83 ± 0.23 | 1.73 ± 0.17 | 1.70 ± 0.12 | **Condition:** BOOT <NORM ($p = 0.027$) |
| | | 3 km/h<br>4 km/h<br>5 km/h | 1.50 ± 0.13<br>1.72 ± 0.11<br>1.85 ± 0.12 | 1.50 ± 0.08<br>1.70 ± 0.12<br>1.80 ± 0.16 | 1.52 ± 0.19<br>1.63 ± 0.09<br>1.77 ± 0.10 | **Speed:** All speeds different to all other speed ($p < 0.001$) |

PWS = preferred walking speed.

*= discrepancy in main effect of condition between PWS and standardised walking speeds.

main effect of condition for ipsilateral swing time ($F_{2,22}$ = 11.62, $p$ < 0.001), and contralateral step length ($F_{2,22}$ = 3.98, $p$ = 0.033) and stance time ($F_{2,22}$ = 5.23, $p$ = 0.014) (Table 1). There were also significant main effects of speed on all spatiotemporal parameters in both limbs ($F_{\geq 1,12, \geq 12.30}$ ≥ 47.38, $p$ < 0.001) (Table 1).

## Discussion

This study aimed to understand the effects of gait speed on lower-limb mechanical work during walking when wearing a CAM boot. We compared total and joint mechanical work between three footwear conditions (NORM, BOOT, EVEN) whilst walking at a PWS and during standardised walking speeds. Overall, comparisons between conditions were generally similar in the ipsilateral limb during PWS and standardised speed trials, including a reduced $W_{tot}$ and $W_{ankle}$. However, there was a change in $W_{tot}$ and $W_{hip}$ in the contralateral limb between footwear conditions at the standardised speeds, but not at PWS.

Ipsilateral (CAM boot wearing) $W_{tot}$ was lower during BOOT and EVEN, which was expected given the ankle's diminished ability to contribute to overall mechanical work during walking [7]. This was corroborated by a reduction in $W_{ankle}$ in BOOT and EVEN when walking at PWS and at standardised speeds of the ipsilateral limb (Fig 2). Although there were no significant effects of footwear condition on ipsilateral $W_{knee}$ and $W_{hip}$, it is important to note that this was consistent between PWS and standardised speeds (Figs 3 and 4). Additionally, this was also shown when joint work was presented as a proportion of $W_{tot}$ (Figs 2–4), suggesting speed does not change the relative contribution of the knee and hip to total mechanical work in the ipsilateral limb. Although relative ankle joint contribution was lower in BOOT and EVEN, similar trends were observed in PWS and across standardised speeds. However, this was not necessarily the case for the contralateral (non-CAM boot wearing) limb, as $W_{tot}$ was unchanged when participants walked at PWS, but was lower in BOOT during the trials where speed was standardised (Fig 1). This raises an important question for researchers, which is whether gait speed should be standardised or freely chosen by patients when monitoring responses to CAM boot wear or, potentially, other foot orthosis. On one hand, standardising walking speeds allows a more direct comparison between footwear conditions, as the total work requirement to complete one gait cycle is equalised for a given distance. This might explain why contralateral $W_{tot}$ was different between conditions in the standardised speed conditions only. However, PWS allows a more natural and ecologically valid response to CAM boot wear given a primary acute response is a reduction in PWS [5,6], although this does mean subsequent alterations to the overall mechanical demand of a gait cycle. PWS is also more implementable in overground data collection, such as in clinical settings when instrumented treadmills are not available. Future research can be confident that adopting a PWS or standardised speed methodological approach will not affect ipsilateral joint kinetics during CAM boot wear, although comparisons in the contralateral limb are less consistent.

In addition to a reduction in contralateral $W_{tot}$ in BOOT at the standardised speeds, BOOT also reduced $W_{knee}$, although this was also the case at PWS (Fig 3). Interestingly, the standardised speeds also showed EVEN increased $W_{knee}$ compared to NORM, which was not shown at PWS. Similar increases in contralateral $W_{hip}$ were also observed. Even-up walkers, or shoe levellers, have previously been shown to partially mitigate losses in gait speed, total mechanical work, and individual joint contribution at PWS during CAM boot wear [7], although the results here show that this might come at the expense of joints compensating for the restrictions associated with CAM boot ambulation. The hip and knee joints of the contralateral limb are some of the more frequently reported sites of secondary pain following CAM boot wear [10]. Whilst it is not clear why pain in these secondary sites occur, it could be suggested that this is a consequence of the increased force due to a leg length discrepancy

[20], or increased mechanical work due to the compensatory mechanism caused by boot wear. The elevated mechanical demand reported here during EVEN potentially shows that these symptoms could be worsened by the prescription of a shoe leveller, which could affect patient compliance to CAM boot wear. Whilst research directly associating CAM boot wear (with or without a contralateral shoe leveller) is scarce, it is plausible to suggest that physical activity could, over time, lead to overuse injuries that align with patient-reported pain. Further longitudinal research is required to provide evidence for this. That being said, given PWS seemed to mitigate responses for the knee joint at least, this is probably more indicative of what patients do during activities of daily living whilst wearing CAM boots, so should offer a clearer understanding of patient responses.

When joint work was displayed as a proportion of $W_{tot}$, there was generally more agreement between PWS and the standardised speeds when comparing footwear conditions. Therefore, if gait speeds must be standardised, presenting joint work as a proportion of the total mechanical work done by that limb is an appropriate method to replicate joint kinetics during PWS. There were also discrepancies (i.e., differences in outcomes) between PWS and standardised speeds for spatiotemporal gait parameters (Table 1). Discrepancies were shown for contralateral step length, stance time, and swing time, and for ipsilateral swing time and cadence. These discrepancies might partially explain those observed in joint kinetics (e.g., reduced swing time in the ipsilateral leg because of the elevated mass caused by the CAM boot), and once again show that controlling gait speed causes participants to unnaturally respond to CAM boots and even-up walkers, strengthening the argument for using PWS in study designs. Comparing spatiotemporal gait parameters at controlled gait speeds might lead to misinterpretation of responses to CAM boot wear, leading to miscalculations of possible overuse injury risks.

Despite the findings discussed above, this study was not without limitations. Firstly, the participant population in the current study were healthy and free of any lower limb injury or condition that might affect gait. Whilst this allowed us to investigate the responses to the CAM boot (and even-up walker) without the confounding effects of a specific injury, it should be acknowledged that responses in a patient population might be different. Future research should consider the findings regarding walking speeds when investigating compensatory strategies during CAM boot wear in patients who require it (e.g., AT rupture patients, patients living with diabetes). The acute responses to CAM boot wear investigated here do not account for any longer-term adaptations, whether that be a learning effect (i.e., finding an optimal compensatory response) or chronic overloading, leading to a worsening of symptoms. Although participants were given two minutes of familiarisation in each CAM boot condition, it is unclear how long the responses presented here last. As such, future research should investigate biomechanical responses to CAM boot wear over a longer time period. The CAM boot used in the current study (Rebound® Air Walker, Össur, Iceland) is only one of many models commonly used in the management of foot and ankle injuries. This CAM boot is a fixed ankle boot, theoretically restricting any ankle motion. In AT rupture patients, these boots would also be fitted with heel wedges, depending on the stage of rehabilitation (the ankle is often gradually moved from an equinus position towards plantigrade over a number of weeks). The configuration used in the current study is therefore indictive of a generic response to CAM boot wear and is not specific to any application or stage of rehabilitation. Rehabilitation management strategies are moving towards more "functional" ankle motion boots, using hinges to control the ankle through a defined range of motion (e.g., VACOped®, Oped GmbH, Germany), depending on the stage of rehabilitation [21,22]. This controlled range of motion might also lead to favourable biomechanical responses at neighbouring joints and in the contralateral limb, so further comparisons are warranted. Finally, individual responses to

CAM boot wear were not explored here, and might, at least in-part, explain some cases where there appear mean differences between conditions that were not statistically significant (e.g., Fig 3B). Inter-individual variability in compensatory strategies during both BOOT and EVEN might not necessarily be articulated by group comparisons. Other confounding factors, such as an individual's muscle strength or variability in natural gait patterns, might also mean some individuals respond differently to others. This should be acknowledged when interpreting comparisons between conditions. Future research might take an individualised approach to understanding acute and chronic responses, which can help tailor pain management strategies and further develop our understanding of secondary site complications.

## Conclusions

Our understanding of the acute biomechanical responses to CAM boot wear can be altered depending on our approach to data collection (PWS or standardised walking speeds). As a result, these differences observed between walking speeds are prudent for future studies to maintain ecological validity by using PWS. Discrepancies in joint work caused by changes in PWS between CAM boot conditions could potentially be offset by normalising to total mechanical work done, which is perhaps the best approach when studying a patient population and should be investigated further.

However, it is also important to consider the potential secondary consequences of CAM boot wear. Ambulation during CAM boot wear, and the implementation of a contralateral even-up walker, can lead to increased mechanical demand on neighbouring joints, which might lead to secondary site complications following prolonged use. Clinicians should consider these biomechanical trade-offs when prescribing CAM boots and consider interventions to mitigate excessive joint loading.

## Supporting information

**S1 Table. Data corresponded to manuscript Figures 1–4.** Total mechanical work and joint mechanical work for the ankle, knee, and hip at each speed and in each footwear condition. ANOVA statistics presented in right-hand column.
(DOCX)

## Acknowledgments

The authors would like to acknowledge Mason L. Stolycia and Samuel Dickinson for their contributions towards data collection.

## Author contributions

**Conceptualization:** Aaron Thomas, David E. Lunn, Josh Walker.

**Data curation:** Aaron Thomas, David E. Lunn, Josh Walker.

**Formal analysis:** Aaron Thomas, David E. Lunn, Josh Walker.

**Investigation:** Aaron Thomas, David E. Lunn, Josh Walker.

**Methodology:** Aaron Thomas, David E. Lunn, Josh Walker.

**Project administration:** Josh Walker.

**Resources:** Aaron Thomas, Josh Walker.

**Software:** Aaron Thomas, David E. Lunn, Josh Walker.

**Supervision:** Josh Walker.

**Validation:** Aaron Thomas, David E. Lunn, Josh Walker.

**Visualization:** Josh Walker.

**Writing – original draft:** Aaron Thomas, David E. Lunn, Josh Walker.

**Writing – review & editing:** Aaron Thomas, David E. Lunn, Josh Walker.

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
