## [Editor Report · Decision Letter 0]

6 Nov 2024

PONE-D-24-25662Gait speed affects hip and knee joint mechanical work done whilst wearing a controlled ankle motion (CAM) bootPLOS ONE

Dear Dr. Walker,

Thank you for submitting your manuscript to PLOS ONE. After careful consideration, we feel that it has merit but does not fully meet PLOS ONE’s publication criteria as it currently stands. Therefore, we invite you to submit a revised version of the manuscript that addresses the points raised during the review process.

 <h3>Major Comments</h3>**Clarity of hypotheses and objectives:**It would benefit the reader to have clearer articulation of the hypotheses and objectives. While the article implies an interest in understanding how mechanical work distribution changes with footwear, explicitly stating the primary hypotheses would enhance the clarity of the research aim.**Explanation of Conditions (NORM, BOOT, EVEN):**The footwear conditions are referenced using abbreviations (NORM, BOOT, EVEN), but the rationale for choosing these conditions is not entirely clear. Please clarify the differences between the BOOT and EVEN conditions and how these comparisons address the study’s primary research question.**Interpretation of Findings in Contralateral Limb:**In the results section, the contralateral limb shows a higher relative knee contribution in the EVEN condition compared to BOOT and NORM, with significant main effects of speed for both limbs. However, the clinical implications of these findings are not fully explored in the discussion. Consider addressing whether increased contralateral knee work could pose risks for injury or discomfort in individuals using CAM boots.**Spatiotemporal Parameters Section:**The section on spatiotemporal parameters presents several effects of condition and speed on variables such as stance time, swing time, and step length. To improve coherence, consider linking these findings more explicitly to the main objectives of the study. Are these changes expected when wearing a CAM boot, and how do they contribute to compensatory strategies?**Potential Confounding Variables:**The study could benefit from acknowledging potential confounding factors that may influence mechanical work distribution, such as participant variability in strength, walking patterns, or experience wearing CAM boots. Addressing these would add depth to the interpretation of the findings. <h3>Minor Comments</h3>**Abbreviations:**Certain abbreviations (e.g., PWS for preferred walking speed) are used inconsistently throughout the text. Ensure all abbreviations are defined upon their first usage and maintain consistency.**Statistical Notation:**The notation for statistical values should be standardized across the article. For example, clarify whether you consistently use one decimal point for p-values (e.g., p < 0.001 instead of p = 0.001) and check that all reported F-values include the correct degrees of freedom.**Figures and Tables Formatting:**Ensure that each figure caption (e.g., Figure 3, Figure 4) briefly summarizes the main result rather than reiterating the same information as in the text. This would improve readability for readers who may reference the figures directly.

We look forward to receiving your revised manuscript.

Kind regards,

Opeyemi Oluwasanmi Adeloye

Academic Editor

PLOS ONE
---

## [Author Response · Author response to Decision Letter 0]

4 Dec 2024

Gait speed affects hip and knee joint mechanical work done whilst wearing a controlled ankle motion (CAM) boot

Author Responses to Reviewer Comments

We would like to thank you for your constructive feedback on our manuscript. We value your comments and suggestions and believe there were in many cases valid. We have acknowledged all comments/recommendations and have offered our responses below in red text. Hopefully, our responses and subsequent manuscript are sufficient to ensure our paper is now acceptable. We feel that these modifications have strengthened the manuscript overall.

Major Comments

Clarity of hypotheses and objectives: It would benefit the reader to have clearer articulation of the hypotheses and objectives. While the article implies an interest in understanding how mechanical work distribution changes with footwear, explicitly stating the primary hypotheses would enhance the clarity of the research aim.

Thank you for this comment. We have amended the final paragraph of our introduction to more clearly articulate the primary hypothesis in our study.

Explanation of Conditions (NORM, BOOT, EVEN): The footwear conditions are referenced using abbreviations (NORM, BOOT, EVEN), but the rationale for choosing these conditions is not entirely clear. Please clarify the differences between the BOOT and EVEN conditions and how these comparisons address the study’s primary research question.

Thank you for this comment. We have added some clarity to the second paragraph of our Introduction to state more clearly what the purpose of an even-up walker is. We have also added two sentences to the end of the Introduction to clarify why including EVEN as a footwear condition is important and clinically relevant, as well as what we might expect given the known impact even-up walkers have on preferred walking speed. Finally, some additional justification has been added to the ‘Data collection’ section of the Materials and Methods.

Interpretation of Findings in Contralateral Limb: In the results section, the contralateral limb shows a higher relative knee contribution in the EVEN condition compared to BOOT and NORM, with significant main effects of speed for both limbs. However, the clinical implications of these findings are not fully explored in the discussion. Consider addressing whether increased contralateral knee work could pose risks for injury or discomfort in individuals using CAM boots.

Thank you for this comment. We agree that this consideration is important. We feel we already discussed the elevated knee joint’s contribution to total mechanical work, and it being a possible explanation for the knee joint being a commonly reported site of pain, in the third paragraph of the discussion. However, we have tried to strengthen this point slightly by discussing the fact that this increased contribution could, over time, pose an injury risk, although we should be wary of too much speculation. We hope that our additional discussion provides more clarity here.

Spatiotemporal Parameters Section: The section on spatiotemporal parameters presents several effects of condition and speed on variables such as stance time, swing time, and step length. To improve coherence, consider linking these findings more explicitly to the main objectives of the study. Are these changes expected when wearing a CAM boot, and how do they contribute to compensatory strategies?

Thank you. We have now referred specifically to spatiotemporal parameters in the final paragraph of the introduction, and we have also developed our interpretation in the discussion. We feel that we did already discuss spatiotemporal data in accordance with the aim of our study somewhat, so have added to this by making a clearer link between these and joint kinetics. We have also discussed how the disagreement in spatiotemporal responses between PWS and standardised speeds is a possible risk for interpretation, especially when discussing the possibility of overuse injuries.

Potential Confounding Variables: The study could benefit from acknowledging potential confounding factors that may influence mechanical work distribution, such as participant variability in strength, walking patterns, or experience wearing CAM boots. Addressing these would add depth to the interpretation of the findings.

Thank you. This suggestion has been implemented in the limitations paragraph of the discussion.

Minor Comments

Abbreviations: Certain abbreviations (e.g., PWS for preferred walking speed) are used inconsistently throughout the text. Ensure all abbreviations are defined upon their first usage and maintain consistency.

Thank you for this comment. We noticed a couple of occasions where Achilles tendon had been spelled out despite us offering an abbreviation (AT) in the opening paragraph of the introduction. This has now been amended on all occasions. The other instances are the use of the abbreviation “CAM” for controlled ankle motion boots, which we have defined in both the abstract and in the introduction (we understand that this is appropriate and standard practice) and “PWS” for preferred walking speed, which we have now also abbreviated in the abstract for consistency. We also amended one occasion in the main text of the manuscript.

Statistical Notation: The notation for statistical values should be standardized across the article. For example, clarify whether you consistently use one decimal point for p-values (e.g., p < 0.001 instead of p = 0.001) and check that all reported F-values include the correct degrees of freedom.

Our p-values were always reported to 3 decimal places (e.g., p = 0.010 or p < 0.001). The cases where we present p < 0.001 are cases where this value falls beyond 3 decimal places in the direction of zero (although p can never = 0). We understand that it is standard practice to present p-values this way. Degrees of freedom have also been added to each instance where an ANOVA F-statistic is presented.

Figures and Tables Formatting: Ensure that each figure caption (e.g., Figure 3, Figure 4) briefly summarizes the main result rather than reiterating the same information as in the text. This would improve readability for readers who may reference the figures directly.

Thank you. We have tried to summarise some main findings from each figure/table in their respective captions. Hopefully this improves readability.

---

## [Decision Letter · Decision Letter 1]

28 Jan 2025

PONE-D-24-25662R1Gait speed affects hip and knee joint mechanical work done whilst wearing a controlled ankle motion (CAM) bootPLOS ONE

Dear Dr. Walker,

Thank you for submitting your manuscript to PLOS ONE. After careful consideration, we feel that it has merit but does not fully meet PLOS ONE’s publication criteria as it currently stands. Therefore, we invite you to submit a revised version of the manuscript that addresses the points raised during the review process.

We look forward to receiving your revised manuscript.

Kind regards,

Daniel J Glassbrook

Academic Editor

PLOS ONE

Journal Requirements:

Additional Editor Comments:

Dear Dr Walker,

Thank you for submitting your revised manuscript. The manuscript is close to being ready for acceptance, however, the reviewers have some minor comments for you to address prior to acceptance of your manuscript.

Best wishes,

Dr Daniel Glassbrook

Reviewers' comments:

Reviewer's Responses to Questions

**Comments to the Author**

1. If the authors have adequately addressed your comments raised in a previous round of review and you feel that this manuscript is now acceptable for publication, you may indicate that here to bypass the “Comments to the Author” section, enter your conflict of interest statement in the “Confidential to Editor” section, and submit your "Accept" recommendation.

Reviewer #1: All comments have been addressed

Reviewer #2: (No Response)

2. Is the manuscript technically sound, and do the data support the conclusions?

Reviewer #1: Partly

Reviewer #2: Yes

3. Has the statistical analysis been performed appropriately and rigorously? 

Reviewer #1: Yes

Reviewer #2: Yes

4. Have the authors made all data underlying the findings in their manuscript fully available?

Reviewer #1: Yes

Reviewer #2: No

5. Is the manuscript presented in an intelligible fashion and written in standard English?

Reviewer #1: Yes

Reviewer #2: Yes

6. Review Comments to the Author

Reviewer #1: This manuscript aims to compare the acute biomechanical response to wearing a CAM boot when walking is freely selected and controlled. Although the research design itself is ambitious, the large number of parameters makes the results difficult to understand.

In addition, it is difficult to understand where there are significant differences, making the results difficult to interpret. Therefore, I think that the results will be easier to understand if they are summarized concisely, such as in a table.

The title does not seem to reflect the content of the study. This manuscript reports that not only changes in walking speed, but also correction of leg length differences using shoe levelers have a significant effect on joint movement when wearing CAM boots.

L33-35

This sentence makes the intent of the manuscript unclear. To resolve this issue, it would be a good idea to explain the advantages of standardizing and measuring speed and the advantages of measuring with PWS, or to mention points to be aware of when comparing standardized speed and PWS.

L.96

The Evenup Shoelift is intended to correct leg length discrepancies when wearing CAM boots, it would be better to label it as "Leg Length Correction (EVAN)."

L101

Please provide details about the specific setup of the CAM boot. For example, was the ankle joint fully fixed or did it have some freedom of movement? Was inversion restricted?

Result

As mentioned at the beginning, I think it will be easier to understand if you delete the results that are not part of your discussion and summarize them concisely, such as in a table.

L292.

This statement does not seem to reflect the results.

It says that there are differences in footwear conditions for Wtot and Whip, but referring to the Figure 3, it seems that there are also differences in Wknee. Also, although it says that there are no significant differences in PWS, there seem to be systematic differences depending on the speed.

L298.

This sentence does not make sense unless it is clearly stated whether it is contralateral or ipsilateral.

In the contralateral position, there is a decrease in Wtot and Whip, but in the ipsilateral position, it is actually higher (especially at 3km/h).

This tendency is also seen in the Wknee, why is this not mentioned?

L325-

It has been described that wearing shoe levelers increases the workload of the opposite knee, causing secondary knee pain, but according to reference 10, secondary knee pain often occurs on the ipsilateral side of the CAM boot, so this logic is incorrect.

Furthermore, it cannot be said with certainty that secondary site pain is caused by compensatory movement, but rather it may be caused by an imbalance in the load due to leg length discrepancy. For this reason, leg length correction should not be easily described as leading to secondary pain.

Minor comments

L26-28. The order of the footwear conditions in L26-28 is reversed from that in L94-96.

Figure legend

The abbreviation TMW should be added to the legend of Fig1-4.

Table

The Left and Right of the Table should also have contralateral and ipsilateral added in parentheses, just like the Figure.

Please add a marker in the table for the discrepancy mentioned in L339.

Reviewer #2: This paper analyzes biomechanical differences in participants walking in CAM boots at a preferred walking speed and standardized walking speeds. One thing I found missing from the introduction was how do the authors think their results should be used in future clinical work. Do they expect their findings to be implemented to analyze CAM boot design or track rehabilitation?

Provide the model (including relevant citation) used to calculate kinematics and kinetics.

Figures. It would be helpful to include some indication of statistical significance on the figures. It was harder to follow having to read back and forth between the figures and results.

Lines 322-323: “Even-up walkers, or shoe levellers, have previously been shown to partially mitigate losses in PWS during CAM boot wear”. Does losses in this case refer to total joint work?

I think the conclusion would read a lot cleaner with a clear take-home message. Based on the authors’ results, should preferred walking speed or normalized speed be used? This would pull the paper together better in the context of their study question.

7. PLOS authors have the option to publish the peer review history of their article (what does this mean? ). If published, this will include your full peer review and any attached files.

**Do you want your identity to be public for this peer review?** For information about this choice, including consent withdrawal, please see our Privacy Policy .

Reviewer #1: No

Reviewer #2: **Yes: ** Karen Kruger

---

## [Author Response · Author response to Decision Letter 1]

12 Mar 2025

We would like to thank you for your constructive feedback on our manuscript. We value your comments and suggestions and believe there were in many cases valid. We have acknowledged all comments/recommendations and have offered our responses below in red text. Hopefully, our responses and subsequent manuscript are sufficient to ensure our paper is now acceptable. We feel that these modifications have strengthened the manuscript overall.

Reviewer #1: This manuscript aims to compare the acute biomechanical response to wearing a CAM boot when walking is freely selected and controlled. Although the research design itself is ambitious, the large number of parameters makes the results difficult to understand.

In addition, it is difficult to understand where there are significant differences, making the results difficult to interpret. Therefore, I think that the results will be easier to understand if they are summarized concisely, such as in a table.

Thank you for this comment, we do agree a table would offer some clarity and we have added a table with the key results as Supplementary Material. This will ensure the focus is still on the key results of the study but will still provide the required clarification.

The title does not seem to reflect the content of the study. This manuscript reports that not only changes in walking speed, but also correction of leg length differences using shoe levelers have a significant effect on joint movement when wearing CAM boots.

Thank you for your comments, and we agree the title doesn’t cover the entirety of the study and have adapted it to reflect the inclusion of the even-up walker. “Exploring mechanical work changes in controlled ankle motion (CAM) boot walking: the effects of gait speed and shoe levelling”

L33-35

This sentence makes the intent of the manuscript unclear. To resolve this issue, it would be a good idea to explain the advantages of standardizing and measuring speed and the advantages of measuring with PWS, or to mention points to be aware of when comparing standardized speed and PWS.

Thank you for this comment. We have added the following text to ensure there is some clarity offered to make a clear recommendation for future studies in the abstract: “Based on these differences observed between walking speeds, it would be prudent for future studies to try to maintain ecological validity by using PWS.” However, we do not think the abstract was necessarily the most appropriate place to discuss the merits of standardising gait speed vs. PWS. As such, we have added the following in the introduction: “To combat this, standardising gait speed across all participants is possible, but lacks ecological validity and patients might adapt to a prescribed walking speed differently, thus altering normal function. Using PWS presents a different challenge when comparing across different participants, because walking speed alters spatiotemporal, kinematic, and kinetic gait parameters (11). To explore how participants adapt during these different conditions…”

L.96

The Evenup Shoelift is intended to correct leg length discrepancies when wearing CAM boots, it would be better to label it as "Leg Length Correction (EVAN)."

Thank you for your comment and we agree this needs more clarity. We have changed this to read “wearing the CAM boot with a leg length corrector (Evenup Shoelift™, Oped GmbH, Germany) on the contralateral left leg (EVEN)”

L101

Please provide details about the specific setup of the CAM boot. For example, was the ankle joint fully fixed or did it have some freedom of movement? Was inversion restricted?

Thank you and this is important information we should have included. We have added the following to the methodological description: “The BOOT condition reflects what a patient who has suffered a lower-leg injury like AT rupture would be prescribed towards the end of rehabilitation, whilst EVEN includes the even-up walker for cases where this is offered to patients. In both BOOT and EVEN, the CAM boot was setup to create a “neutral”, plantigrade ankle position (i.e., no plantarflexion or dorsiflexion), and no internal ankle range of motion (in any plane) was, in theory, permitted.”

Result

As mentioned at the beginning, I think it will be easier to understand if you delete the results that are not part of your discussion and summarize them concisely, such as in a table.

We have responded to this comment earlier and have added Supplementary Table (Supplementary Table S1).

L292.

This statement does not seem to reflect the results.

It says that there are differences in footwear conditions for Wtot and Whip, but referring to the Figure 3, it seems that there are also differences in Wknee. Also, although it says that there are no significant differences in PWS, there seem to be systematic differences depending on the speed.

Thank you for this comment. Whilst we do note that there appear to be some differences for Wknee, these were not significant. Therefore, we do not feel it is appropriate to speculate on this occasion. This absence of “statistical significance” where there appears to be an effect could be because of inter-subject variability in responses (i.e., the between-subject variability exceeds the between-condition differences). This is a point we addressed initially in our limitations section, but we have now added to it to hopefully provide more clarity about this point in particular: “and might, at least in-part, explain some cases where there appear mean differences between conditions that were not statistically significant (e.g., Figure 3B).”

L298.

This sentence does not make sense unless it is clearly stated whether it is contralateral or ipsilateral.

We have added the ipsilateral limb to line 315 (in the latest version) as this was the only place we could see it was missing, line 298 (in the previous version) appeared to be complete as the study’s aim was to understand the effects of gait speed and the shoe leveller on lower-limb mechanical work for both limbs.

In the contralateral position, there is a decrease in Wtot and Whip, but in the ipsilateral position, it is actually higher (especially at 3km/h).

This tendency is also seen in the Wknee, why is this not mentioned?

Whilst we agree with these observations, as mentioned above, we only reported and discussed statistically significant findings.

L325-

It has been described that wearing shoe levelers increases the workload of the opposite knee, causing secondary knee pain, but according to reference 10, secondary knee pain often occurs on the ipsilateral side of the CAM boot, so this logic is incorrect.

Furthermore, it cannot be said with certainty that secondary site pain is caused by compensatory movement, but rather it may be caused by an imbalance in the load due to leg length discrepancy. For this reason, leg length correction should not be easily described as leading to secondary pain.

Thank you for this comment, whilst you are correct the secondary site pain was more common in the ipsilateral knee (10/31 patients), pain in the contralateral knee was still observed in a similar number of patients (8/31). We have added an additional sentence to address this: “Whilst it is not clear why pain in these secondary sites occur, it could be suggested that this is a consequence of the increased force due to a leg length discrepancy (Li et al 2015) or increased mechanical work due to the compensatory mechanism caused by boot wear. The…”

Minor comments

L26-28. The order of the footwear conditions in L26-28 is reversed from that in L94-96.

Thank you we have changed the order in the abstract.

Figure legend

The abbreviation TMW should be added to the legend of Fig1-4.

Thank you we have now added this.

Table

The Left and Right of the Table should also have contralateral and ipsilateral added in parentheses, just like the Figure.

Thank you this has been done for this table and the added table with the results included.

Please add a marker in the table for the discrepancy mentioned in L339.

We have amended Table 1 by adding asterisks whenever there is a discrepancy between the main effects of condition for PWS and for standardised speeds. For consistency, we have also done this in the new Supplementary Table S1.

Reviewer #2: This paper analyzes biomechanical differences in participants walking in CAM boots at a preferred walking speed and standardized walking speeds. One thing I found missing from the introduction was how do the authors think their results should be used in future clinical work. Do they expect their findings to be implemented to analyze CAM boot design or track rehabilitation?

Thank you for this comment. The aim of the work was two-fold. Firstly, to understand how changes in gait speed affect mechanical work of the lower limb joints. This was primarily to understand whether future studies, or where treadmill walking is used as a form of rehabilitation, should use a standardised gait speed or preferred walking speed, or indeed if it even matters in terms of patient function. Secondly, it was to understand the effect of wearing an Even-up walker. Depending on the results this could add to the evidence of why a shoe levelling device (a.k.a. leg length corrector) should be used by, or prescribed for, patients when using a CAM boot.

To address the clarity around the aim of the shoe levelling device, we have added the following to lines 83-84 of the latest manuscript version: "The results of this study could inform the importance of prescribing a shoe-leveller device for patients using a CAM boot.”

We have also strengthened the rationale for the use of preferred walking speed and standardised speeds in the introduction, now including: “To combat this, standardising gait speed across all participants is possible, but lacks ecological validity and patients might adapt to a prescribed walking speed differently, thus altering normal function. Using PWS presents a different challenge when comparing across different participants, because walking speed alters spatiotemporal, kinematic, and kinetic gait parameters (11). To explore how participants adapt during these different conditions…”

Provide the model (including relevant citation) used to calculate kinematics and kinetics.

Thank you for this comment and sorry this was not clear in our previous manuscript. We have amended the relevant sentence in the methods to now read as: “Retroreflective markers were placed according to the calibrated anatomical systems technique (CAST), where the locations of skin-based joint markers have been described previously (7, 15).”

Figures. It would be helpful to include some indication of statistical significance on the figures. It was harder to follow having to read back and forth between the figures and results.

Thank you, for this comment and we agree it would make it easier for the reader to follow. However, a number of attempts, adding indicators of statistical significance to the figures made them look untidy and probably less readable. This is because of the multiple speeds and condition (several within-subjects factors and interactions). However, we have now included a data table in the Supplementary Material (Supplementary Table S1), which was suggested by the other review. This table contains the information from the ANOVA associated with each of the figures in the manuscript. Hopefully, this provides clarity here.

Lines 322-323: “Even-up walkers, or shoe levellers, have previously been shown to partially mitigate losses in PWS during CAM boot wear”. Does losses in this case refer to total joint work?

Thank you for the comments, and we agree this sentence isn’t as clear as it could be. In terms of losses, we are referring to changes in total joint work and individual joint contributions as well as actual gait speed itself. We have amended the sentence accordingly: “Even-up walkers, or shoe levellers, have previously been shown to partially mitigate losses in gait speed, total mechanical work, and individual joint contribution at PWS during CAM boot wear”

I think the conclusion would read a lot cleaner with a clear take-home message. Based on the authors’ results, should preferred walking speed or normalized speed be used? This would pull the paper together better in the context of their study question.

Thank you for this comment, and we have adapted our conclusion to make a clear take-home message from the PWS vs standardised walking speed perspective and the use of an even-up walker: “Our understanding of the acute biomechanical responses to CAM boot wear can be altered depending on our approach to data collection (PWS or standardised walking speeds). As a result, these differences observed between walking speeds are prudent for future studies to maintain ecological validity by using PWS. Discrepancies in joint work caused by changes in PWS between CAM boot conditions could potentially be offset by normalising to total mechanical work done, which is perhaps the best approach when studying a patient population and should be investigated further.

However, it is also important to consider the potential secondary consequences of CAM boot wear. Ambulation during CAM boot wear, and the implementation of a contralateral even-up walker, can lead to increased mechanical demand on neighbouring joints, which might lead to secondary site complications following prolonged use. Clinicians should consider these biomechanical trade-offs when prescribing CAM boots and consider interventions to mitigate excessive joint loading.”

---

## [Editor Report · Decision Letter 2]

14 Mar 2025

Exploring mechanical work changes in controlled ankle motion (CAM) boot walking: the effects of gait speed and shoe levelling

PONE-D-24-25662R2

Dear Dr. Walker,

We’re pleased to inform you that your manuscript has been judged scientifically suitable for publication and will be formally accepted for publication once it meets all outstanding technical requirements.

Kind regards,

Daniel J. Glassbrook

Academic Editor

PLOS ONE
---

## [Editor Report · Acceptance letter]

PONE-D-24-25662R2

PLOS ONE

Dear Dr. Walker,

I'm pleased to inform you that your manuscript has been deemed suitable for publication in PLOS ONE. Congratulations! Your manuscript is now being handed over to our production team.

Kind regards,

on behalf of

Dr. Daniel J. Glassbrook

Academic Editor

PLOS ONE